# SARS-CoV-2 Variants of Concern: Presumptive Identification via Sanger Sequencing Analysis of the Receptor Binding Domain (RBD) Region of the *S* Gene

**DOI:** 10.3390/diagnostics13071256

**Published:** 2023-03-27

**Authors:** Grazielle Motta Rodrigues, Fabiana Caroline Zempulski Volpato, Priscila Lamb Wink, Rodrigo Minuto Paiva, Afonso Luís Barth, Fernanda de-Paris

**Affiliations:** 1Residência Multiprofissional em Saúde e em Área Profissional da Saúde do Hospital de Clínicas de Porto Alegre, Porto Alegre 90035-903, Rio Grande do Sul, Brazil; 2Serviço de Diagnóstico Laboratorial, Hospital de Clínicas de Porto Alegre, Porto Alegre 90035-903, Rio Grande do Sul, Brazil; 3LABRESIS–Laboratório de Pesquisa em Resistência Bacteriana, Hospital de Clínicas de Porto Alegre, Porto Alegre 90035-903, Rio Grande do Sul, Brazil; 4Programa de Pós-Graduação em Ciências Farmacêuticas, Universidade Federal do Rio Grande do Sul, Porto Alegre 90160-093, Rio Grande do Sul, Brazil

**Keywords:** SARS-CoV-2, Sanger sequencing, RBD, VOCs

## Abstract

Variants of concern (VOCs) of SARS-CoV-2 are viral strains that have mutations associated with increased transmissibility and/or increased virulence, and their main mutations are in the receptor binding domain (RBD) region of the viral spike. This study aimed to characterize SARS-CoV-2 VOCs via Sanger sequencing of the RBD region and compare the results with data obtained via whole genome sequencing (WGS). Clinical samples (oro/nasopharyngeal) with positive RT-qPCR results for SARS-CoV-2 were used in this study. The viral RNA from SARS-CoV-2 was extracted and a PCR fragment of 1006 base pairs was submitted for Sanger sequencing. The results of the Sanger sequencing were compared to the lineage assigned by WGS using next-generation sequencing (NGS) techniques. A total of 37 specimens were sequenced via WGS, and classified as: VOC gamma (8); delta (7); omicron (10), with 3 omicron specimens classified as the BQ.1 subvariant and 12 specimens classified as non-VOC variants. The results of the partial Sanger sequencing presented as 100% in agreement with the WGS. The Sanger protocol made it possible to characterize the main SARS-CoV-2 VOCs currently circulating in Brazil through partial Sanger sequencing of the RBD region of the viral spike. Therefore, the sequencing of the RBD region is a fast and cost-effective laboratory tool for clinical and epidemiological use in the genomic surveillance of SARS-CoV-2.

## 1. Introduction

SARS-CoV-2′s rapid worldwide spread has caused the emergence of new lineages of great importance for the pandemic scenario. The new lineages may present an accelerated rate of transmission which results in a continuous and rapid process of emergence of other new mutant variants [1]. Until now, there were five SARS-CoV-2 variants of concern (VOCs) as defined by the World Health Organization (WHO); these variants have mutations with functional significance in the *S* gene in common, responsible for expressing the glycoprotein spike [2,3]. 

Based on the sequences deposited in the National Genomics Data Center (NGDC) of China, more than 27,500 mutations in the S gene have already been identified and documented which may cause changes in some sites of the spike protein amino acid sequence. Considering these genomic alterations, more than 7000 mutations cause some alteration in the sequence of the receptor binding domain (RBD) of the spike protein [4]. The significance of the accumulation of mutations in the spike protein, especially in the RBD region, is due to the important role that this region plays in the main process of virus entrance into the host cells. In fact, the entrance process is mediated by the affinity of the SARS-CoV-2 RBD region with the angiotensin-converting enzyme 2 receptor (ACE2). The RBD is also important due to the fact that it is the main target of neutralizing antibodies. Therefore, the fixation of mutations in the spike protein allows an important advantage in the virus’s replication rate and is usually used for variant determination [5,6].

The evaluation of SARS-CoV-2 mutations and their lineage determination is carried out via whole genome sequencing (WGS) using next-generation sequencing (NGS) methods, which allows for phylogenetic assignment [1,7]. The NGS gives accurate information of genetic variability of the virus, as it is able to generate comprehensive genomic data which allows for tracing of the origin and spread of the virus, besides monitoring its evolution [8]. However, WGS is a time-consuming assay, the data analysis is complex, and it is an expensive technique, especially in resource-limited settings. In this sense, it is important to evaluate alternative techniques for identifying SARS-CoV-2 variants and carrying out genomic surveillance of VOCs [9]. The Sanger sequencing technique is considered the standard method for short nucleotide sequence determination, it is available to many labs, it is less expensive and faster than NGS, and it can run samples individually [10]. These features allow Sanger sequencing to be used as a screening method for detecting SARS-CoV-2 mutations and generating data of importance for public health and surveillance systems [10,11].

The VOCs carry their main mutations in the RBD region of the spike protein, which indicates that a technique capable of predicting VOCs based on the mutational profile of this region could contribute to genomic surveillance [5]. In this context, we proposed a simplified Sanger sequencing assay of the RBD region in the *S* gene to presumptively characterize all SARS-CoV-2 variants of concern described until now. 

## 2. Materials and Methods

### 2.1. Clinical Specimens and Ethical Statement

Thirty-seven oro/nasopharyngeal swabs with positive results for SARS-CoV-2 according to the RT-qPCR protocol contained in the Centers for Disease Control and Prevention (CDC) guidelines were included in this study [12]. As part of a genomic surveillance research, all of the RNA sequences from the clinical specimens were submitted for whole genome sequencing (WGS). This study was approved by the Ethics Committee of the Hospital de Clínicas de Porto Alegre, CAAE number: 48879321000005327.

### 2.2. In Silico Analysis

In order to evaluate the capability for distinguishing between VOCs and non-VOCs sequences using only the RDB region analysis, we use an in silico approach. Sequences from alpha, beta, gamma, delta, and omicron VOCs and some omicron subvariants (BA.1, BA.2, BA.4, BA.5 and BQ.1) were downloaded from the GISAID Database. Sequences were selected based on whether their status was both “complete” (>29,000 nucleotides) and classified as “high coverage” (<0.05% of unique amino acid mutations). These sequences were aligned in the BioEdit and CodonCode Aligner software programs to the SARS-CoV-2 reference with the names NC_045512.2 (complete genome), NC_045512.2:21563-25384 (gene *S*), and NC_045512.2:22517-23522 (representing the 1006 bp fragment amplified via PCR). 

### 2.3. RNA Extraction and cDNA Synthesis

The RNA was extracted from the clinical samples using the commercially available QIAamp Viral RNA Kit (Qiagen, Valencia, CA, USA) according to the manufacturer’s instructions. Reverse transcription was performed using GoScript™ Reverse Transcriptase (Promega, Madison, WI, USA) with an optimized half-reaction: 0.5 μL of a random primer was incubated with 2 μL of RNA at 70 °C for 5 min and was afterwards quickly chilled on ice for 5 min. This mixture was added to a reverse transcription mix containing 2 μL of GoScript™ 5X Reaction Buffer, 0.6 μL of MgCl2 (25 mM), 0.5 μL of PCR Nucleotide Mix, 3.65 μL of Nuclease-Free Water, 0.25 μL of Recombinant RNasin^®^ Ribonuclease Inhibitor, and 0.5 of μL GoScript™ Reverse Transcriptase. The product, which had a final volume of 10 μL, was submitted to a temperature of 25 °C for 5 min (annealing), followed by 42 °C for 60 min (cDNA synthesis), and then was heated to 70 °C for 15 min (inactivation of reverse transcriptase). 

### 2.4. RBD Polymerase Chain Reaction (PCR) Amplification and Sanger Sequencing

The PCR for the RBD region was performed using the primers 75L (5′-AGAGTCCAACCAACAGAATCTATTGT-3′) and 77R (5′-CAGCCCCTATTAAACAGCCTGC-3′) designed by ARTIC protocol [13]. The predicted PCR amplicon with these primers is a 1006 bp product flanking the RBD region of the spike protein of the SARS-CoV-2 virus. The PCR was prepared as described by Dorlas et al. [11], and the products were analyzed in 1% agarose gel electrophoresis (40 min at 110 v). The products were purified with ExoSAP-IT^TM^ PCR Product Cleanup (Affymetrix, Santa Clara, CA, USA) according to the manufacturer’s protocol. The final product was used in the Sanger sequencing that was carried out with the BigDye™ Terminator v3.1 Cycle Sequencing Kit (Applied Biosystems, Foster City, CA, USA). The sequencing was processed in an ABI 3500 Genetic Analyzer (Applied Biosystems, Foster City, CA, USA). 

### 2.5. Limit of Detection (LoD) and Repeatability

To determine the limit of detection, we evaluated a serial dilution (from 1:1 to 1:10,000) of SARS-CoV-2 positive samples in a RT-qPCR assay and used a standard curve to quantify the viral load, as previously described by Wink et al. [14]. The same dilutions of SARS-CoV-2 positive specimens were submitted for RBD PCR. The final dilution determined as the LoD was submitted a series of 20 parallel PCRs to establish repeatability.

### 2.6. Bi-Directional Sanger Sequencing Analysis

The data obtained from Sanger sequencing were aligned with the SARS-CoV-2 reference sequence (NC_045512.2) and gene *S* (NC_045512.2:21563-25384) using the ClustalW multiple method with BioEdit Alignment Editor software v.7.2 and CodonCode Aligner v.10 software. The quality of the sequencing data was assessed using Sanger electropherograms of both forward and reverse sequences. The prediction of VOCs was evaluated according to the presence/absence of SNVs (single nucleotide variants) in comparison with the reference sequence (Table 1). The pairwise sequence alignment score was obtained using the ClustalW multiple method using BioEdit software. The mutated base quality was analyzed using the Phred quality values, calculated using CodonCode Aligner software, that represent the probability of error for each base call. The quality values analyzed considered the Phred score for both the forward and the reverse sequences and the consensus scores for the two sequences.

## 3. Results

### 3.1. In Silico Analysis

In the in silico analysis, all VOCs could be differentiated based on the 75L/77R 1006 bp fragment (Figure 1). Furthermore, the omicron subvariants BA.2 and BQ.1 could be distinguished from omicron BA.1. The BA.4 and BA.5 omicron subvariants could not be differentiated based on their 75L/77R fragments. This fragment comprises the mutations present between nucleotides 22,517 and 23,521 of the SARS-CoV-2 genome. All VOCs’ mutations and omicron subvariants’ mutations investigated in this nucleotide interval are listed in Table 1 and Table 2, respectively, along with the main established nomenclature systems for the VOCs. 

### 3.2. Sanger Sequencing and WGS Results

We successfully sequenced the RBD region from the 37 clinical samples using the Sanger sequencing technique. It was possible to identify the gamma (8/37), delta (7/37), and omicron (10/37) VOCs, while 12 samples were classified as non-VOCs. Moreover, among the omicron variants, it was also possible to identify the subvariant BQ.1 (3/10). Among the non-VOCs sequenced, it was possible to differentiate the zeta (5/12) and lambda (1/12) variants of interest (VOI). It was not possible to distinguish the other non-VOC linages by sequencing the 75L/77R 1006 bp fragment. The prediction of SARS-CoV-2 VOC and non-VOCs using Sanger sequencing presented as 100% in concordance with the results generated from WGS. The SNVs identified and the presumptive categorization from the 37 clinical samples and the lineage assigned via WGS are shown in Table 3. 

### 3.3. Sanger Sequencing Quality

We found that the 1006 bp amplicon was able to cover the entire RBD region with a high degree of quality in the Sanger sequencing results when we performed the bi-directional analysis. The protocol described in this study was able to generate fragments with an average of 886 and 867 bases of the forward and reverse sequences, respectively, with a high degree of quality. Analysis of the alignment with the reference sequence showed that the average percentage of matches was 95% for the forward sequence and 93% for the reverse sequence. The average of the quality of the base calls was 52 for the forward sequences and 56 for the reverse sequences, which means that there was only about a one in 100,000 chance that the base call was incorrect. When we analyzed the forward and reverse sequences together (bi-directional sequencing), the base call quality consensus increased to 88, which means that there was about a one in 100,000,000 chance that the base call was incorrect, demonstrating the high quality and fidelity of the base call. It was possible to obtain sequencing results using the Sanger technique with only the forward or the reverse sense for seven of the 37 specimens, even when the procedure was repeated. Although these samples gave rise to only one sequencing sense, it was possible to identify a VOC along the sequence with a high base call quality score. 

### 3.4. Limit of Detection

The limit of detection for the RBD region generated via PCR was determined as shown in Table 4. The lower viral load limit for performing the RBD PCR was around 500,000 copies/uL (dilution 1:100). This viral load corresponds to a cycle threshold (Ct) of around 20 in the RT-qPCR from the CDC protocol. All specimens included in this study had a RT-qPCR Ct value, as defined in the CDC protocol for detecting SARS-CoV-2, of less than 20, and specimens that had Ct values higher than 20 were not amplified using the RBD PCR protocol. At these lower copy detection limits, a repeatability assay of 20 parallel RBD PCRs was performed and 95% (19/20) of these were amplified. 

## 4. Discussion

In this study, we proposed an approach that allows for the performance of genomic surveillance of VOCs based on an analysis of the RBD region in the S gene of SARS-CoV-2 using partial and bi-directional Sanger sequencing. We found that the Sanger sequencing results of the 75L/77R 1006 bp fragment presented as 100% in agreement with WGS for lineage determination. This protocol was initially developed when only the alpha (B.1.1.7), beta (B.1.351), and gamma (P.1) variants were circulating around the world. Later on, the protocol was applied to also detect the delta (B.1.617.2) and omicron (B.1.1.529) variants, including the omicron subvariant BQ.1. In this study, we show that it is possible to identify these VOCs and differentiate between them and differentiate them from non-VOCs using only one PCR fragment. 

The Sanger protocol was also able to identify some non-VOC variants by confirming the absence of the mutations of concern described in Table 1 or by confirming the presence of only D614G mutations in the analyzed region. The protocol was also able, without the need for extra adaptation, to detect additional VOCs not described when the technique was originally developed, showing that this method can be used as a generic approach to target specific mutations to distinguish other potential VOCs that may appear in the future. Although the VOCs share some identical mutations, each variant has a unique combination of mutations which generates a specific mutational profile in the RBD region [15,16]. 

The RBD region is composed of 749 nucleotides and the concentration of the lineage-defining mutations in this region allows for analysis via Sanger sequencing, which supports the generation of sequences up to 1000 bp [17,18]. In general, longer fragments are challenging to use due the difficulty of using Sanger sequencing to distinguish single base pair differences at the end of fragments up to 900 bp long and the loss of the first 15–40 bases due to primer binding [17]. Bi-directional sequencing helps to improve the analysis efficiency for longer fragments, such as 75L/77R, and enables the analysis of mutations that are located at the beginning or at the end of the fragment with a higher degree of quality. Among the mutations identified via Sanger sequencing in the RBD region, the G22578A mutation which leads to the spike G339D mutation in the omicron variant is located at the beginning of the fragment used in our study. In all omicron samples it was possible to identify this mutation, but the maximum base call quality score obtained for the consensus sequences was 47. Despite the fact that there is a low probability of an incorrect base call, this low score indicates that this mutation is located in a fragment region that has a lower quality of sequencing than other regions of the amplicon. However, this low score does not compromise the identification of omicron variant, as its RBD mutational profile is very different from the other VOCs [19].

The unique mutational profile of the omicron variant allows it to be easily distinguished from alpha, beta, gamma, delta, and non-VOC variants, but the emergence of omicron descendents’ lineages makes the differentiation between omicron variants and their subvariants via the RBD region difficult. However, a mutational profile change in the omicron subvariants allows for differentiation of the descendents’ lineages from omicron via the RBD region based on the presence of the A22688G, G22775A, and A22786C mutations (that lead to the T376A, D405N, and R408S spike mutations, respectively). These mutations are absent in the omicron BA.1 variant and are an indication that the sample is an omicron subvariant [20]. In this way, the BA.2 and BQ.1 omicron subvariants that have been circulating the most recently in Brazil can be distinguished via key mutations in the RBD region [21]. Nevertheless, the BA.4 and BA.5 variants cannot be differentiated via only the RBD analysis due their identical spike sequences, as shown in Table 2 [3,20]. In our study, we also sequenced the RBD region from the BQ.1 omicron subvariant, and differentiation between the other subvariants was possible due to the A22893C change (that leads to the spike K444T mutation), which is not present in the omicron variants or in other omicron subvariants.

Due to the large size of the fragment 75L/77R, a high viral load is required to provide reliable results from Sanger sequencing. Based on a comparative analysis, the LOD was determined as a minimal viral load of 500,000 copies/uL using an adapted RT-qPCR CDC protocol [14], corresponding to Ct values of up to 20, the same described in another study [22]. This viral load is necessary to generate a proper sequencing electropherogram that allows for an accurate analysis with no ambiguous base calls. The need for a high viral load seems not to pose a problem, as most of the Ct values in symptomatic SARS-CoV-2-positive patients are lower than 20, especially when the clinical samples are collected within 10 days since symptom onset [23,24]. Moreover, the high viral load does not prevent performance of the test, as greater risk of transmission, as in an outbreak scenario, is associated with increased viral load values and positive relationships between viral load and infectiousness [25].

The Sanger sequencing protocol proposed in this study is able to sequence a 1006 bp fragment using a bi-directional approach with high accuracy to distinguish among VOCs via analysis of the RBD region of the *S* gene from the SARS-CoV-2 virus, generating a result in 100% agreement with the lineage definition generated via WGS. This approach allows for the identification of mutational profiles of SARS-CoV-2 VOCs from an individual sample with a lower time burden and a lower cost in comparison with the WGS techniques. Of note is the fact that this is possible using a unique PCR fragment [11]. Considering the emergence of new VOCs with mutations in the RBD portion, this protocol can be applied to predict VOCs and discriminate amongst them. Hence, Sanger sequencing can be used as an important tool for the screening and identification of VOCs to provide data for the genomic surveillance of SARS-CoV-2. Furthermore, clinical assistance could be improved, as the rapid results of Sanger sequencing is useful for the differentiation between re-infection versus persistent infections in SARS-CoV-2-positive patients during prolonged periods, for example.

## 5. Conclusions

The fast and accessible determination of VOCs is an essential tool for SARS-CoV-2 genomic surveillance. In this study, Sanger sequencing of the RBD region was shown to be in agreement with SARS-Cov-2 lineage assigned via WGS. This analysis of the RBD Sanger sequencing was capable of detecting all five VOCs already described by the WHO. Therefore, Sanger sequencing of the RBD region is a potential applicable and cost-effective laboratory tool for clinical and epidemiological use in the genomic surveillance of SARS-CoV-2.

## Figures and Tables

**Figure 1 diagnostics-13-01256-f001:**
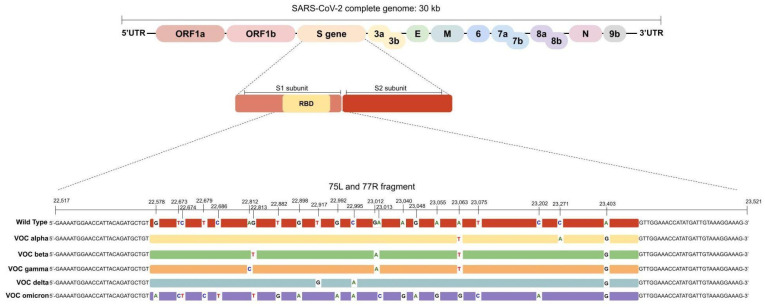
Genomic organization of SARS-CoV-2. The RBD region is located inside the S1 subunit from the S gene. The 75L/77R fragment comprises all mutations of concern from the VOCs along the RBD region. The wild type sequence refers to the SARS-CoV-2 reference sequence NC_045512.2.

**Table 1 diagnostics-13-01256-t001:** Summary of genomic annotation, amino acid changes, and the main established nomenclature systems for the SARS-CoV-2 variants of concern.

WHO Variant of Concern	Genomic Annotation	Amino Acid Change	Pangolineage	Nextstrain Clade
Alpha	A23063T	N501Y	B.1.1.7	20I (V1)
	C23271A	A570D		
	A23403G	D614G		
Beta	G22813T	K417N	B.1.351	20H (V2)
	G23012A	E484K		
	A23063T	N501Y		
	A23403G	D614G		
Gamma	A22812C	K417T	P.1	20J (V3)
	G23012A	E484K		
	A23063T	N501Y		
	A23403G	D614G		
Delta	T22917G	L452R	B.1.617.2	21A, 21I, 21J
	C22995A	T478K		
	A23403G	D614G		
Omicron	G22578A	G339D	B.1.1.529	21K, 21L, 21M
	T22673C	S371L		
	C22674T			
	T22679C	S373P		
	C22686T	S375F		
	G22813T	K417N		
	T22882G	N440K		
	G22898A	G446S		
	G22992A	S477N		
	C22995A	T478K		
	A23013C	E484A		
	A23040G	Q493R		
	G23048A	G496S		
	A23055G	Q498R		
	A23063T	N501Y		
	T23075C	Y505H		
	C23202A	T547K		
	A23403G	D614G		

**Table 2 diagnostics-13-01256-t002:** Summary with genome annotation and amino acid changes in omicron subvariants BA.1, BA.2, BA.4, BA.5, and BQ.1.

BA.1 Omicron Descendent Lineage		BA.2 Omicron Descendent Lineage		BA.4 and 5 Omicron Descendent Lineage		BQ.1 Omicron Descendent Lineage	
Genome Annotation	Amino Acid Change	Genome Annotation	Amino Acid Change	Genome Annotation	Amino Acid Change	Genome Annotation	Amino Acid Change
G22578A	G339D	G22578A	G339D	G22578A	G339D	G22578A	G339D
-	R346T absent	-	R346T absent	-	R346T absent	G22599C	R346T
-	L368I absent	-	L368I absent	-	L368I absent	-	L368I absent
T22673C/ C22674T	S371L	C22674T	S371F	C22674T	S371F	C22674T	S371F
T22679C	S373P	T22679C	S373P	T22679C	S373P	T22679C	S373P
C22686T	S375F	C22686T	S375F	C22686T	S375F	C22686T	S375F
-	T376A absent	A22688G	T376A	A22688G	T376A	A22688G	T376A
-	D405N absent	G22775A	D405N	G22775A	D405N	G22775A	D405N
-	R408S absent	A22786C	R408S	A22786C	R408S	A22786C	R408S
G22813T	K417N	G22813T	K417N	G22813T	K417N	G22813T	K417N
T22882G	N440K	T22882G	N440K	T22882G	N440K	T22882G	N440K
-	K444T absent	-	K444T absent	-	K444T absent	A22893C	K444T
-	V445P absent	-	V445P absent	-	V445P absent	-	V445P absent
G22898A	G446S	-	G446S absent	-	G446S absent	-	G446S absent
-	L452R/Q absent	-	L452R/Q absent	T22917G	L452R	T22917G	L452R
-	N460K absent	-	N460K absent	-	N460K absent	T22942A	N460K
G22992A	S477N	G22992A	S477N	G22992A	S477N	G22992A	S477N
C22995A	T478K	C22995A	T478K	C22995A	T478K	C22995A	T478K
A23013C	E484A	A23013C	E484A	A23013C	E484A	A23013C	E484A
-	F486V/S absent	-	F486V/S absent	T23018G	F486V	T23018G	F486V
-	F490S absent	-	F490S absent	-	F490S absent	-	F490S absent
A23040G	Q493R	A23040G	Q493R	-	Q493R absent	-	Q493R absent
G23048A	G496S	-	G496S absent	-	G496S absent	-	G496S absent
A23055G	Q498R	A23055G	Q498R	A23055G	Q498R	A23055G	Q498R
A23063T	N501Y	A23063T	N501Y	A23063T	N501Y	A23063T	N501Y
T23075C	Y505H	T23075C	Y505H	T23075C	Y505H	T23075C	Y505H
C23202A	T547K	-	T547K absent	-	T547K absent	-	T547K absent
A23403G	D614G	A23403G	D614G	A23403G	D614G	A23403G	D614G

**Table 3 diagnostics-13-01256-t003:** Clinical samples sequenced for presumptive variant categorization and the previous lineage assigned via WGS, Ct in the RT-qPCR, fragment size obtained including quality, and the pairwise alignment score compared to the reference sequence.

	CDC RT-qPCR Ct		Fragment Size (bp)		Pairwise Sequence Alignment Score		SNVs Identified via Sanger Sequencing	Sanger Presumptive Identification	Lineage Assigned via WGS	
Sample ID	N1	N2	FWD	REV	FWD	REV			PangoLineage	WHO VOCs
1	14.99	15.16	922	ND	98%	ND	A22812C; G23012A; A23063T; A23403G	gamma	P.1	gamma
4	14.86	15.17	904	838	95%	89%	A23403G	non-VOC	B.1.1.28	non-VOC
5	16.67	15.28	858	817	92%	92%	A22812C; G23012A; A23063T; A23403G	gamma	P.1	gamma
6	15.04	15.48	898	868	97%	93%	G23012A; A23403G	non-VOC/zeta	P.2	non-VOC/zeta
8	15.26	14.15	837	781	94%	94%	G23012A; A23403G	non-VOC/zeta	P.2	non-VOC/zeta
11	16.48	16.56	898	848	95%	91%	T22917A; T23031C; A23403G	non-VOC/lambda	C.37	non-VOC/lambda
12	13.18	13.85	910	865	96%	95%	A22812C; G23012A; A23063T; A23403G	gamma	P.1	gamma
13	14.97	13.94	941	932	97%	99%	T22917A; C22995A; A23403G	delta	B.1.617.2	delta
14	18.32	18.57	936	920	97%	97%	T22917A; C22995A; A23403G	delta	B.617.2-like	delta
15	12.03	12.41	933	921	98%	98%	A23403G	não-VOC	B.1.1.29	non-VOC
22	14.44	15.24	929	934	97%	96%	A22812C; G23012A; A23063T; A23403G	gamma	P.1	gamma
23	18.71	17.65	914	896	96%	94%	A22812C; G23012A; A23063T; A23403G	gamma	P.1	gamma
30	13.29	12.2	928	920	98%	95%	T22917A; C22995A; A23403G	delta	B.617.2-like	delta
32	15.94	15.1	898	868	97%	93%	G23012A; A23403G	non-VOC/zeta	P.2	non-VOC/zeta
33	12.11	12.03	887	890	95%	95%	G23012A; A23403G	non-VOC/zeta	P.2	non-VOC/zeta
34	18.18	18.48	885	792	95%	89%	A22812C; G23012A; A23063T; A23403G	gamma	P.1	gamma
35	13.65	13.41	886	850	93%	93%	A23403G	non-VOC	B.1.1.28	non-VOC
39	20.16	19.14	897	ND	96%	ND	-	non-VOC	B.1.1.28	non-VOC
40	18.47	19.71	768	494	95%	91%	A23403G	non-VOC	B.1.1.28	non-VOC
42	18.67	19.99	826	771	94%	93%	A22812C; G23012A; A23063T; A23403G	gamma	P.1	gamma
47	13.48	14.15	893	898	96%	94%	A22812C; G23012A; A23063T; A23403G	gamma	P.1	gamma
51	12	12.5	877	810	94%	92%	G22578A; T22673C; C22674T; T22679C; C22686T; G22813T; T22882G; G22898A; G22992A; C22995A; A23013C; A23040G; G23048A; A23055G; A23063T; T23075C; C23202A; A23403G	omicron	B.1.1.529	omicron
52	13	14.3	877	867	92%	93%	G22578A; T22673C; C22674T; T22679C; C22686T; G22813T; T22882G; G22898A; G22992A; C22995A; A23013C; A23040G; G23048A; A23055G; A23063T; T23075C; C23202A; A23403G	omicron	B.1.1.529	omicron
58	10.98	9.18	877	867	95%	95%	G23012A; A23403G	non-VOC/zeta	P.2	non-VOC/zeta
59	19.15	19.32	705	837	92%	94%	A23403G	non-VOC	B.1.1.28	non-VOC
60	18.08	19.32	886	889	94%	94%	G22578A; T22673C; C22674T; T22679C; C22686T; G22813T; T22882G; G22898A; G22992A; C22995A; A23013C; A23040G; G23048A; A23055G; A23063T; T23075C; C23202A; A23403G	omicron	B.1.1.529	omicron
61	20.87	32.82	854	837	92%	92%	G22578A; T22673C; C22674T; T22679C; C22686T; G22813T; T22882G; G22898A; G22992A; C22995A; A23013C; A23040G; G23048A; A23055G; A23063T; T23075C; C23202A; A23403G	omicron	B.1.1.529	omicron
62	23.79	29.02	879	369	95%	93%	T22917A; C22995A; A23403G	delta	B.617.2-like	delta
63	19.89	20.4	ND	692	ND	94%	T22917A; C22995A; A23403G	delta	B.617.2-like	delta
64	19.86	30.81	880	831	93%	92%	G22578A; T22673C; C22674T; T22679C; C22686T; G22813T; T22882G; G22898A; G22992A; C22995A; A23013C; A23040G; G23048A; A23055G; A23063T; T23075C; C23202A; A23403G	omicron	B.1.1.529	omicron
65	12.97	12.89	928	900	96%	95%	T22917A; C22995A; A23403G	delta	B.617.2-like	delta
66	17.38	22.78	839	ND	93%	ND	G22578A; T22673C; C22674T; T22679C; C22686T; G22813T; T22882G; G22898A; G22992A; C22995A; A23013C; A23040G; G23048A; A23055G; A23063T; T23075C; C23202A; A23403G	omicron	B.1.1.529	omicron
68	16.05	20.8	901	873	92%	93%	G22578A; T22673C; C22674T; T22679C; C22686T; G22813T; T22882G; G22898A; G22992A; C22995A; A23013C; A23040G; G23048A; A23055G; A23063T; T23075C; C23202A; A23403G	omicron	B.1.1.529	omicron
69	19.14	29	879	890	95%	93%	T22917A; C22995A; A23403G	delta	B.617.2-like	delta
70	ND	ND	897	ND	93%	ND	G22599C, C22674T, T22679C, C22686T, A22688G, G22775A, A22786C, G22813T, T22882G, A22893C, T22917G, T22942A, G22992A, C22995A, A23013C, T23018G, A23055G, A23063T, T23075C, A23403G	omicron, sublinage BQ.1	BQ.1	omicron, sublinage BQ.1
73	ND	ND	876	ND	94%	ND	G22599C, C22674T, T22679C, C22686T, A22688G, G22775A, A22786C, G22813T, T22882G, A22893C, T22917G, T22942A, G22992A, C22995A, A23013C, T23018G, A23055G, A23063T, T23075C, A23403G	omicron, sublinage BQ.1	BQ.1	omicron, sublinage BQ.1
74	ND	ND	760	ND	91%	ND	G22599C, C22674T, T22679C, C22686T, A22688G, G22775A, A22786C, G22813T, T22882G, A22893C, T22917G, T22942A, G22992A, C22995A, A23013C, T23018G, A23055G, A23063T, T23075C	omicron, sublinage BQ.1	BQ.1	omicron, sublinage BQ.1

Ct—cycle threshold; bp—base pairs; N1—nucleocapsid 1; N2—nucleocapsid 2; FWD—forward sense; REV—reverse sense; ND—not determined.

**Table 4 diagnostics-13-01256-t004:** Standard curve quantification for determining the sample minimum viral load limit of detection.

Serial Dilution	Target	Ct Standard	VL Standard (Copies/uL)	Ct Sample	VLSample (Copies/uL)	RBD PCR Result
1:1	N1	23.41	100,000	13.28	60,740,000	Positive
	N2	23.56	100,000	15.04	15,915,000	
1:10	N1	26.86	10,000	17.25	4,709,500	Positive
	N2	26.87	10,000	18.48	1,918,500	
1:100	N1	30.1	1000	20.62	540,550	Positive
	N2	30.53	1000	22.4	171,750	
1:1000	N1	33.77	100	25.88	18,285	Negative
	N2	34.29	100	27.55	7252	
1:10,000	N1	38.18	10	30.9	742	Negative
	N2	38.57	10	33.11	245	

Ct—cycle threshold; VL—viral load; N1—nucleocapsid 1; N2—nucleocapsid 2.

## Data Availability

The raw data supporting the findings of this study are available from the corresponding author on reasonable request, including the raw sequencing data. All other data generated and analyzed during the realization of this study are included in this paper.

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
