# Peer review of "SARS-CoV-2 Variants of Concern: Presumptive Identification via Sanger Sequencing Analysis of the Receptor Binding Domain (RBD) Region of the S Gene"

_diagnostics, 2023, doi:10.3390/diagnostics13071256_

Round 1

Reviewer 1 Report

Journal: Diagnostics (ISSN 2075-4418)

Manuscript ID: diagnostics-2149677

Type: Brief Report

Title: SARS-CoV-2 variants of concern: presumptive identification by Sanger sequencing analysis of the receptor binding domain (RBD) region of the S gene

Results section:

The results paragraphs should be restructured to improve the reader's comprehension. Also leave more space between paragraphs and tables.

Author Response

Response to Reviewer 1 Comments

Point 1: Results section: The results paragraphs should be restructured to improve the reader's comprehension.

Response 1: We understand the concern of the reviewer. However, in this situation, the results sections are constructed in a logical order, to facilitate the comprehension of the reader. We maintained the same order, although we created subheadings to separate the topics. 

Point 2: Also leave more space between paragraphs and tables.

Response 2: We thank the reviewer for his/her positive feedback on our manuscript. It was probably a formatting error and we already corrected it.

Reviewer 2 Report

It would be appropriate to explain the term "in silico".

It is recommended to add literature to the material and method section.

Author Response

Response to Reviewer 2 Comments

Point 1: It would be appropriate to explain the term "in silico".

Response 1: We thank the reviewer for his/her suggestion. We rewrote the first line of item "2.2. In silico analysis" to improve the comprehension about the in silico analysis that was performed.

The first part of our experiment was performed using software in order to simulate the analysis of the Sanger sequencing of the RBD region, to verify if it was possible to distinguish the VOCs among themselves and among other non-VOCs. After we verified that it was possible to distinguish, we performed the sequencing of the samples that we had available.

Point 2: It is recommended to add literature to the material and method section.

Response 2: We agree with the reviewer that it should be added. We add literature to the material and method section to give the credits to the Arctic Network, which provided a list of primers to sequencing the entire genome of SARS-CoV-2 and whose we choose the 75L and 77R primers used in our study.

Reviewer 3 Report

The purpose, methodology and clinical usefulness of this novel approach to an important clinical problem appeared to me to be sound.  It would appear to be a clinically and epidemiologically useful novel methodology that may prove helpful in the fight against the ongoing Covid pandemic.  The English was excellent and required little or no editing.

Author Response

Response to Reviewer 3 Comments

Point 1: The purpose, methodology and clinical usefulness of this novel approach to an important clinical problem appeared to me to be sound.  It would appear to be a clinically and epidemiologically useful novel methodology that may prove helpful in the fight against the ongoing Covid pandemic.  The English was excellent and required little or no editing.

Response 1: Dear reviewer, we are glad with your comments about the paper. We propose this methodology in order to collaborate with genomic surveillance with a useful tool and less sophisticated than NGS to allow the identification of the variants already described.

It may be potentially applicable and cost-effective laboratory tools in a scenario of limited resources. In this way, we show that by a partial sequencing, using a part of the SARS-CoV-2 genome that is under selective pressure and contains the main defining lineage's mutations, it is possible to identify the VOCs and differentiate them from the non-VOCs.

Reviewer 4 Report

The manuscript “SARS-CoV-2 variants of concern: presumptive identification by Sanger sequencing analysis of the receptor binding domain (RBD) region of the S gene” offers an alternative of a sequencing method which could be used in places where more sofisticated equipment isnt available

General comments and questions

The authors said that they use a pair of primers that amplifies a region in RBD of about 1006 bp (line 89), but I did a quit primer blast analysis in it gave a differentsize amplicon, about 600 pb, so please explain if you are using a less astringent bioinformatic tool, or why is your amplicon different to what is expected in silico?

Minor revisions:

1) line 84 has a typo error between BA.5 and BQ.1, correct “e” for “and”

2) line 136, why BA.1 is in parenthesis and also in line 227

Author Response

Response to Reviewer 4 Comments

Point 1: The manuscript “SARS-CoV-2 variants of concern: presumptive identification by Sanger sequencing analysis of the receptor binding domain (RBD) region of the S gene” offers an alternative of a sequencing method which could be used in places where more sophisticated equipment isn't available.

General comments and questions

The authors said that they use a pair of primers that amplifies a region in RBD of about 1006 bp (line 89), but I did a quit primer blast analysis in it gave a different size amplicon, about 600 pb, so please explain if you are using a less astringent bioinformatic tool, or why is your amplicon different to what is expected in silico?

Response 1:

The primers are selected from a list of primers available by the Arctic Network, as mentioned in the text. This initiative designed a list of primers that cover the entire genome of SARS-CoV-2 in order to facilitate the efforts to sequencing the virus during the beginning of the pandemic. In this way, we did not design primers.

We thank the reviewer for showing us the difference of amplicon' size. In order to check your information, we revised and corrected the sequences of primers in the text. Primers 75 L (5’-AGAGTCCAACCAACAGAATCTATTGT-3') stars in the nucleotide position 22,517 and primer 77R (5’-CAGCCCCTATTAAACAGCCTGC-3') end in the nucleotide position 23,522, forming a fragment of 1,006 base pairs

Point 2: Minor revisions:

1) line 84 has a typo error between BA.5 and BQ.1, correct “e” for “and”

2) line 136, why BA.1 is in parenthesis and also in line 227

Response 2: Thank you so much for the comments. These observations were corrected in the manuscript.

Reviewer 5 Report

In the era of -omics and emergence of new SARS-CoV-2 genetic variants, we quite often focus on the sophisticated methods of analysis and often forget about the older but still good methodology. Motta and colleagues in their manuscript compared the next generation sequencing with Sanger sequencing to identify new SARS-CoV-2 variants of concern. In my opinion, the paper is well-written, and conclusions are clear. However, I have only one minor comment, which may increase the quality of the paper.

1.     The Sanger sequencing has some limitations, i.e. the sensitivity, and the 1000 bp fragment may be a bit problematic and it is possible that some mutations may be omitted. Did authors concern to use for example 3 pair of primers, which would result in three 300-400 bp fragments for sequencing? The smaller fragments gives higher chance that some rare mutations will be identified.

Author Response

Response to Reviewer 5 Comments

Point 1: In the era of -omics and emergence of new SARS-CoV-2 genetic variants, we quite often focus on the sophisticated methods of analysis and often forget about the older but still good methodology. Motta and colleagues in their manuscript compared the next generation sequencing with Sanger sequencing to identify new SARS-CoV-2 variants of concern. In my opinion, the paper is well-written, and conclusions are clear. However, I have only one minor comment, which may increase the quality of the paper.

  1. The Sanger sequencing has some limitations, i.e. the sensitivity, and the 1000 bp fragment may be a bit problematic and it is possible that some mutations may be omitted. Did authors concern to use for example 3 pair of primers, which would result in three 300-400 bp fragments for sequencing? The smaller fragments gives higher chance that some rare mutations will be identified.

Response 1:

We thank the reviewer's comments. We understand your point, but it is not exactly a limitation of our study design. We also tested including more primers to improve the sensitivity, but the cost would increase due to the greater number of reactions required and this point is a bigger limiter than the sensitivity in scenarios of resources limited, as the Brazilian public health system. We bet on this only fragment in order to facilitate the analysis and generate lower costs to define the variant of concern, once we sequence only one fragment.

In addition, minor fragments allow that  samples with lower viral load can be amplified with better quality to sequence. But, when we started this study, the variant gamma was emergent in our country, and we needed to differentiate the gamma variant from others non-VOCs in our institution. With the introduction of the delta variant in Brazil, we use the same protocol. In this moment, we saw that this only fragment could be an important tool to differentiate the VOCs. The same occurred with the introduction of omicron and their subvariants. So, using a part of the SARS-CoV-2 genome that is under selective pressure and contains the main defining lineage's mutations, it is possible to identify the VOCs and differentiate them from the non-VOCs.

So, in order to clarify this point in the reading, we put the sentence below in the first paragraph of the discussion section.

"In this study, we show that it is possible to identify the VOCs and differentiate between themselves and from the non-VOCs using only one PCR fragment."

Reviewer 6 Report

1: This MS lacks novel findings. Many papers have been published similar to this article.

2: Why not XBB.1.5, XBB and BA.2.75 used for comparison?

3: No negative control for Ct value.

4: Why the sequences were used from Brazil, authors should use from other locations also.

5: Limit of detection is not a parameter for sequence variation analysis.

6:Even less copy number of virus may result approximaely 20 Ct value in RT-qPCR.

7:Authors stated that this is low cost and less time consuming but it is not like that, this system is costly and time consuming.

8: There are some mutations have also been reported in SI, not only in RBD, so it is not useful always to decide the genome variaton and emrgnce of new VOC.

9:I did not find either in the introduction or discussion about what are the other genes that have mutations and what is their role in the emergence of VOC/VOI.

10: There are many other genes including structural and non-structural proteins that have some mutations and many reviews and papers have been published related to this information.

11: Based on the overall data provided and the final conclusion, this MS can not be accepted for publication.

Author Response

Response to Reviewer 6 Comments

Point 1:  This MS lacks novel findings. Many papers have been published similar to this article.

Response 1: We understand your point. But when we started this study, we had an increased number of cases in our country due to the introduction of the gamma variant and we needed to differentiate the gamma variant from other non-VOCs in our institution.

Some published articles bring Sanger sequencing as an alternative to the NGS to identify the variant of concern, but always proposing a series of fragments so that VOCs can be clearly differentiated. This is a limitation in countries with resource-limited to the public health system.  So, we show that using a part of the SARS-CoV-2 genome which is under selective pressure and contains the main defining lineage's mutations, it is possible to identify the VOCs and differentiate them from the non-VOCs.

With this same protocol, without any changes with the emerging to the new VOCs, we were able to differentiate the gamma from non-VOCs and delta, omicron and even some subvariants of omicron, following the changes in the RBD region of SARS-CoV-2.

Point 2:  Why not XBB.1.5, XBB and BA.2.75 used for comparison?

Response 2:  This paper was created during the pandemic of SARS-CoV 2, with the rapid emergence of variants of concern (Gamma, Delta and Omicron). Indeed, these omicron subvariants did not included in the paper. With the rapid emergence of omicron subvariants and the need to differentiate from omicron, we included in the analysis the subvariants. Although in our local epidemiology, in southern Brazil, until the moment the subvariants XBB.1.5, XBB and BA.2.75 did not emerge in our region.

Point 3:  No negative control for Ct value.

Response 3: During the PCR amplification, in all the batches are included the CN and CP control. Although, we did not use a negative control for the SARS-CoV-2 genome. We analyze only positive samples in the RT-qPCR for the targets N1 and N2 (as recommended by CDC). Our objective is to compare the information that we can extract from the Sanger sequence by looking at only a portion of the SARS-CoV-2 genome rather than the information given by the WGS.

Point 4:  Why the sequences were used from Brazil, authors should use from other locations also.

Response 4: As mentioned in the response 2, we are located in southern Brazil.

The first step of our work (in silico analysis) was verified if it was possible to distinguish the VOCs among themselves. To do this, we selected sequences deposited in the GISAID, from many countries, to evaluate if it was possible to distinguish the VOCs among themselves and among other non-VOCs with the sequence that we choose. It proved feasible to VOCs around the world.

Nonetheless, when we did the "in vitro'' step, we used the specimens that were accessible for us. As mentioned we are from Brazil and this paper was done in a COVID-19 reference tertiary-care hospital. Even more, we had access to WGS results that were helpful to validate our results.

Point 5:  Limit of detection is not a parameter for sequence variation analysis.

Point 6: Even less copy number of virus may result approximaely 20 Ct value in RT-qPCR.

Response to points 5 and 6:  Actually, we use the limit of detection as a parameter of sample selection to analyze by Sanger sequencing, not as a sequence variation parameter. We perceived that when we have a lower viral load, with less than approximately 500,000 copies (as shown in table 4 of the manuscript) the PCR amplification is not satisfactory and the Sanger technique showed poor quality. The quality of Sanger sequencing is an important parameter because a poor sequence can lead to an error in the base call or in the operator interpretation in the analysis. To eliminate this variable, we put this limit of detection as a "cut off" point in the PCR step. Furthermore, we determine which Ct we have this limited viral load for the PCR step, using a quantification standard curve with a commercial positive control with determined viral load. We know that the Ct is not a quantitative parameter and precisely by this that we quantify to have this information.

Point 7:  Authors stated that this is low cost and less time consuming but it is not like that, this system is costly and time consuming.

Response 7:  We proposed that the approach using Sanger sequencing can be used as an important tool for screening and identification of VOCs to provide data for genomic surveillance of SARS-CoV-2 in scenarios with no sophisticable tools to genomic determination as NGS.

We are located in a public  COVID-19 reference tertiary-care hospital. In this scenario, we often have to sequence samples individually, and sequence a unique PCR fragment using Sanger technique was low cost and less time consuming when compared to NGS technique.

In order to clarify this point, we rewrite a sentence in the discussion section, as shown below.

This approach allows the identification of mutational profiles of SARS-CoV-2 VOCs in an individual sample with less time consuming and a lower cost comparison to the WGS techniques. Noteworthy, this is possible using a unique PCR fragment.”

Point 8:  There are some mutations have also been reported in SI, not only in RBD, so it is not useful always to decide the genome variaton and emrgnce of new VOC.

Response 8: We thank the reviewer suggestion; however, it is not exactly the objective of the study design. In our study we wanted to show, using RBD, choosing a part of the SARS-CoV-2 genome that is under strong selective pressure and contains the main defining lineage's mutations that allow changes in their transmissibility. So, in this context, using a  partial sequencing we showed that it is possible to identify the VOCs and differentiate them from the non-VOCs.

Point 9: I did not find either in the introduction or discussion about what are the other genes that have mutations and what is their role in the emergence of VOC/VOI.

Point 10: There are many other genes including structural and non-structural proteins that have some mutations and many reviews and papers have been published related to this information.

Response to points 9 and 10: We thank the reviewer comments; however we respectfully disagree. We did not include in our discussion or introduction other genes that are under selective pressure in the virus evolution because the focus of our study is on the RBD region of spike. This region is the most exposed part of the virus in the host environment and submitted to the neutralizing antibodies, and for this reason is under a strong selective pressure.

Round 2

Reviewer 6 Report

I found that authors have responded up to some extent but at this stage, the generated data is not much useful. Much updated information has been published related to this work.